# Gold-Nanoparticle-Coated Magnetic Beads for ALP-Enzyme-Based Electrochemical Immunosensing in Human Plasma

**DOI:** 10.3390/ma15196875

**Published:** 2022-10-03

**Authors:** Seo-Eun Lee, Se-Eun Jeong, Jae-Sang Hong, Hyungsoon Im, Sei-Young Hwang, Jun Kyun Oh, Seong-Eun Kim

**Affiliations:** 1Human IT Convergence Research Center, Convergence System R&D Division, Korea Electronics Technology Institute (KETI), 25 Saenari-ro, Bundang-gu, Seongnam-si 13509, Korea; 2Department of Polymer Science and Engineering, Dankook University, 152 Jukjeon-ro, Suji-gu, Yongin-si 16890, Korea; 3Center for Systems Biology, Massachusetts General Hospital, Boston, MA 02114, USA; 4Department of Radiology, Massachusetts General Hospital, Boston, MA 02114, USA

**Keywords:** gold nanoparticle, magnetic bead, electrochemical immunoassay, indium tin oxide, human plasma

## Abstract

A simple and sensitive AuNP-coated magnetic beads (AMB)-based electrochemical biosensor platform was fabricated for bioassay. In this study, AuNP-conjugated magnetic particles were successfully prepared using biotin–streptavidin conjugation. The morphology and structure of the nanocomplex were characterized by scanning electron microscopy (SEM) with energy-dispersive X-ray analysis (EDX) and UV–visible spectroscopy. Moreover, cyclic voltammetry (CV) was used to investigate the effect of AuNP-MB on alkaline phosphatase (ALP) for electrochemical signal enhancement. An ALP-based electrochemical (EC) immunoassay was performed on the developed AuNP-MB complex with indium tin oxide (ITO) electrodes. Subsequently, the concentration of capture antibodies was well-optimized on the AMB complex via biotin–avidin conjugation. Lastly, the developed AuNP-MB immunoassay platform was verified with extracellular vesicle (EV) detection via immune response by showing the existence of EGFR proteins on glioblastoma multiforme (GBM)-derived EVs (10^8^ particle/mL) spiked in human plasma. Therefore, the signal-enhanced ALP-based EC biosensor on AuNP-MB was favorably utilized as an immunoassay platform, revealing the potential application of biosensors in immunoassays in biological environments.

## 1. Introduction

Biosensors have been intensively studied for their rapid and sensitive techniques that are highly dependent on substrate material [1,2,3]. Among the various signaling techniques for biosensors, electrochemical-based biosensing platforms remain a high priority owing to their low cost and high sensitivity [4,5,6,7,8]. Moreover, magnetic particle allows great flexibility in designing electrochemical biosensors for a wide range of analytes for simple and early disease diagnostics [9,10]. In addition, various nanoparticle strategies have been developed to sensitively detect biomolecules on electrochemical sensor platforms. Here, the selection of the nanoparticles plays a key role in determining the overall sensor performance. Gold nanoparticles (AuNPs) are one of the most promising materials as they provide a highly sensitive technique to detect bioanalytes in biosensors due to their physicochemical properties [11,12,13]. Moreover, the application of AuNP and MB combinations for biosensor development has been a recent focus due to their unique physicochemical properties [14,15,16,17,18].

Over the past few decades, magnetic particles (MPs or MBs) made of iron oxide have been developed in many fields, such as biomedicine, magnetic resonance imaging, and optics [19,20,21,22,23]. MBs have recently attracted more attention in biomedical applications because of their magnetic characteristics, ease of use, and high sensitivity [24,25]. In particular, MBs provide an efficient separation process in bioassay platforms, leading to sample concentration [25,26]. Recently, MBs used as a bioanalyte carrier in the EC-based biosensing platforms have been reported because the enhanced surface area of MBs allows signal amplification as well [27,28,29,30,31,32,33]. However, because MB material generally has low conductivity [34,35], it is not suitable for charge transport in the EC-sensing platforms. To further increase the charge transport and biocompatibility, the surface of the MB generally needs to be modified with some metal or polymer. Gold nanoparticles (AuNPs), a metal with good biocompatibility, are commonly used in biomedical applications. Due to their biocompatibility, gold-coated MPs have been developed and widely studied. Moreover, gold-nanoparticle-coated MBs simultaneously possess magnetic characteristics and charge transport enhancement.

To develop an electrochemical immunoassay platform, indium tin oxide (ITO) was utilized for the sensing electrode owing to its low capacitive background currents and low cost [36,37]. In addition, the alkaline phosphatase (ALP) enzyme used in this study is known as one of the most commonly used enzymes generating electrochemical signaling molecules for electrochemical immunosensors due to its stability and electrochemically proper pair of a substrate and product [38,39,40], such as ascorbic acid-2-phosphate (AAP) and L-ascorbic acid (AA). In biosensor platforms, a nonelectrochemical reaction such as Ab–Ag binding can be converted into an electrochemical signaling process by labeling Ab with alkaline phosphatase (ALP) reacting with AAP substrate, generating an electroactive AA. Lastly, electrochemical signal measurements, such as cyclic voltammetry or chronocoulometry, were performed to obtain the EC signal from the oxidation of AA at the applied potential. In this platform, it has been applied with the AuNP-coated MBs for the bioassay on the ITO electrode for exosome detection. Exosome, known as extracellular vesicles (EV), is a well-known circulating biomarker carrier in extracellular spaces. This is a highly promising protein cargo to examine a variety of diseases in diagnostics and therapeutics [41,42].

In this work, an AuNP-coated MB preparation and its application in bioassay for profiling EV were examined, as shown in Figure 1. We assembled AuNPs on magnetic particles and coated their surfaces with bioreceptors, such as antibodies, via streptavidin or avidin (STV or AV)–biotin conjugation. Streptavidin/avidin is a commonly used biomaterial for developing new biomedical methods due to the high affinity between STV/AV and biotin. We experimentally demonstrated that AuNP-layered MBs were able to improve the electron transfer to the electrode surface in the biosensor platform. Lastly, on the developed AMB-based bioassay platform, GBM-derived EVs spiked in human plasma were successfully detected and profiled with disease-related biomarkers such as EGFR. This result suggested that the AuNP-MB complex is a promising material to exploit in biological assay techniques on magnetic-particle-based electrochemical immunoassay systems for point-of-care diagnostics.

## 2. Materials and Methods

### 2.1. Materials

Avidin, biotin-conjugated gold nanoparticles, alkaline phosphatase-linked anti-mouse IgG, magnesium chloride, Trizma base, bovine serum albumin, and human plasma were purchased from Sigma Aldrich (St. Louis, MO, USA). Pierce^TM^ alkaline phosphatase biotinylated and 20× PBS Tween 20 were purchased from Thermo Scientific (Waltham, MA, USA). Commercial magnetic particles, Dynabeads™ M-280 Streptavidin, were obtained from Invitrogen (Carlsbad, CA, USA). Anti-EGFR or anti-EpCAM antibodies with biotin and anti-CD63 antibodies were purchased from Abcam (Cambridge, UK). Phosphate buffered saline (PBS, 10×, pH 7.4) was obtained from iNtRON (Seongnam, Gyeonggi, Korea). L-ascorbic acid 2-phosphate from Tokyo Chemical Industry (Chuo-ku, Tokyo, Japan) was utilized as a substrate in our study. All the binding steps were processed in pretreated e-tubes with bovine serum albumin and Tween 20. Indium tin oxide (ITO)-deposited transparent glass through sputtering methods was purchased from Buwon-PS (Taoyuan District, Taiwan). Potentiostat, SP-150 from BioLogic (Claix, France), was utilized as an electrochemical signal measuring instrument.

### 2.2. Preparation of AuNP-MB Complex

A total of 2 μL of streptavidin-coated magnetic beads (10 mg/mL) was inserted in e-tubes to bind with 1 μL of biotin-conjugated AuNPs (2.66 mg/mL) for 1 h at room temperature (RT). For stabilizing the mixture on the magnet, removing steps were followed after 10 min. We used 1× PBS for washing steps to remove any unbound AuNPs, and 10 min of stabilization on top of the magnet was necessarily performed to allow the magnetic beads to settle before removing solutions. Avidin (10 μg/mL, 15 μL) was added to the AuNP-MB-generating sites to bind biotin-conjugated biomolecules such as ALP and antibodies. Then, the mixture was placed for 1 h at RT. Then, the washing steps were repeated twice. For the characterization, the synthesized particles were transferred to a 96-well plate, and absorbance spectra were obtained using UV–Vis spectroscopy, FLUOstar Omega from BMG LABTECH (Ortenberg, Germany). The 96-well plates were from SPL life Sciences (Pocheon, Gyeonggi, Korea). All the experiments for the characterization of AuNPs were performed three times. For the field-emission scanning electron microscopy (FE-SEM) with energy-dispersive X-ray analysis (EDX), a JSM-7900F from JEOL Ltd. (Tokyo, Japan) was also used to examine the morphology of the AuNP-MBs (Figure 2 and Appendix A).

### 2.3. Electrochemical Analysis of AuNP-MB Complex Labeled with ALP

Biotinylated ALP was added to the prepared AuNP-MB complex, and then the mixture was incubated for 1 h at room temperature. Then, washing steps were carried out. The AuNP-MB complex labeled with ALP was redispersed with 1× PBS, and the solution was transferred to the magnet-attached ITO electrode. For the ITO electrode, ITO deposited glasses were diced into a 5 × 15 mm rectangular shape. Each diced ITO glass was covered with punched Teflon tape to form a working area of 3 mm diameter circular shape. Lastly, the current responses from cyclic voltammetry (CV) of the samples in 1 mM AAP after 2 min of incubation were measured by the ITO electrodes.

### 2.4. Immunoassays on AuNP-MB Complex for GBM-Derived EV Detection

First, using various concentrations of biotinylated anti-EpCAM antibodies (0.1, 1, 5, or 10 μg/mL) as capture antibodies, the concentration of capture antibodies on AuNP-MB complex was optimized by quantifying with anti-mouse IgG antibodies labeled with ALP (40 μg/mL) (Figure 3). Then, for the EV immunoassay, 15 μL of biotinylated anti-EGFR or anti-EpCAM antibodies (10 μg/mL) diluted in 1× PBS was added into the prepared AuNP-MB complex. After pipetting with the samples for mixing, the mixture was incubated for 30 min at room temperature, and we performed the washing step mentioned earlier. Purified EVs (10^8^ ptcl/mL, Gli36 from GBM cell line), isolated from cell culture as previously described [43], were added and incubated for 1 h at RT. Lastly, alkaline-phosphatase-conjugated anti-CD63 antibodies (20 μg/mL) were bound for 1 h at RT to the capture EVs on the AuNP-MB for the sandwich assay. The quality of the used EV was analyzed in advance with a Malvern NanoSight NS300 from Malvern Panalytical (Malvern, United Kingdom) (Appendix A).

### 2.5. Electrochemical Signal Measurements

The prepared ITO working electrode, and counter and reference electrodes were connected to the potentiostat by alligator clips. For the measurement, Ag/AgCl (3M NaCl) reference and Pt wire counter were used. All three electrodes were settled in a cell and dipped in 1 mL of ascorbic acid phosphate (AAP) substrate. After 2 min for ALP enzyme reaction with AAP, cyclic voltammograms (CV) and chronocoulogram (CC) were measured for each sample.

## 3. Results and Discussions

### 3.1. Characterization of AuNP-MP Complex

To prepare well-dispersed AuNP-coated MBs, we immobilized AuNPs on MBs through streptavidin–biotin conjugation methods. In this study, we utilized streptavidin-coated MB and 20 nm diameter biotinylated gold nanospheres, which are known as reasonably stable, safe, and high-performance for biomedical applications [44,45] (Figure 1). The mixture was incubated at a 1:10 ratio of streptavidin-coated MB to biotinylated AuNP at room temperature, followed by washing steps. In order to examine the morphology and optical property of AuNP-coated MBs, the AuNP-modified MBs were characterized by field-emission scanning electron microscopy (FE-SEM) combined with EDX and UV–Vis spectroscopy (Figure 2).

To investigate the binding between AuNPs and MBs, compared with the MB without AuNPs, the AuNP-coated MBs showed AuNPs (~20 nm in diameter) bound to the external surface of MBs (Figure 2a,b). The size of AuNPs in the SEM image was measured using the xT microscope control program and was around 20 nm in diameter, as expected (Appendix A). In addition, SEM-EDX elemental mapping data showed that the Au was present on MBs (Figure 2c). Thus, we verified that the AuNPs were properly conjugated with the MB through the streptavidin–biotin binding process. Due to its surface plasmon resonance, AuNP shows characteristic absorbance. Thus, biotinylated AuNP shows a UV–visible absorbance peak at 520 nm that is characteristic of its surface plasmon resonance (Figure 2d). As shown in Figure 2d, the UV–Vis spectra showed that the maximum absorbance peak of AuNPs shifted from 520 to 548 nm upon the binding of AuNPs on MBs. None of the AuNPs showed severe spectral changes, i.e., the sign of aggregation. Compared with the total absorbance of MBs, a new absorption maximum at 548 nm in AuNP-bound MBs was also observed. Therefore, the AuNP-coated MB was successfully verified by SEM, EDX, and UV–Vis spectra.

### 3.2. Electrochemical Signal Enhancement of ALP on AuNP-MP Complex

In order to investigate the effect of AuNPs on electrochemical systems, we first prepared an ALP-immobilized ITO electrode surface using streptavidin-labeled ALP with various biotinylated AuNP (~20 nm) concentrations, as shown in Appendix A. For electrochemical measurements, the surface-modified working electrode was placed in 1 mM AAP substrate for 2 min, while AAP was converted into AA, which is an electroactive molecule. Then, cyclic voltammetry (CV) was carried out in the range of −0.3 to 0.9 V to measure the electron transfers from the oxidation of AA at the applied potentials. As shown in Appendix A, the electrochemical signal amplification due to AuNPs was confirmed by AuNP increasing as the current signal at 0.6 V increased as much as 5.23 $\times 10^{-2}$ mA (33.14%) and 6.46$\times 10^{-2}$ mA (64.52%) for 0.5 and 2.5 µL of AuNP, respectively, from the signal of ALP without AuNP (3.93$\times 10^{-2}$ mA). In addition, to examine the signal enhancement effect on the ALP-based electrochemical assay by AuNP-bound MBs, biotinylated ALP-conjugated MBs (MB-ALP), and biotinylated AuNP bound MBs with an additional avidin layer; then biotinylated ALP (MB-AuNP-avidin-ALP) was prepared as described in the Materials and Methods. Additionally, controls (MB and MB-AuNP-avidin without ALP) were prepared as well. As shown in Figure 3b, the bar graph for each case is drawn with the current value at 0.6 V in the graph in Figure 3a. The following graph shows a clear signal difference for each case. The results of this study showed that the current values of MB-ALP and MB-AuNP-avidin-ALP were 1.25$\times 10^{-3}$ and 3.83$\times 10^{-3}$ mA, respectively. As a result, the signal of MB-AuNP-avidin-ALP was amplified by 206.40% compared with that of MB-ALP without AuNP. Therefore, we verified that the signal enhancement in the ALP-based electrochemical assay with MB was significantly affected by AuNP.

### 3.3. Optimized Conditions for Electrochemical Immunoassay on AuNP-MB Complex

To construct an electrochemical biosensor platform based on AuNP-MB, antibodies were immobilized on the surface of the AuNP-MB complex. To conjugate the capture antibodies on the AuNP-MB, the AuNP-MB complex was first coated with avidin through the biotin moiety of AuNPs on MB. Then, biotinylated capture antibodies were bound to the avidin-coated AuNP-MB complex. For the capture antibodies, anti-EpCAM antibodies, one of the disease-related biomarkers on EVs, were utilized to optimize the conditions for antibody immobilization. To analyze the antibodies’ immobilization, anti-mouse IgG antibodies labeled with ALP for EC signals were treated, which could bind to the anti-EpCAM antibodies conjugated on the AuNP-MB because the anti-EpCAM antibodies originated from mice and their isotype were IgG. In Figure 4, as the treated concentration of antibodies on the AuNP-MB complex increased to 10 µg/mL, current signals from ALP enzyme reactions of anti-mouse IgG antibodies bound to anti-EpCAM antibodies on the AuNP-MB gradually increased until antibodies fully covered the surface of AuNP-MB. The conditions for antibodies immobilization were also produced in a biological environment including 90% human plasma, and we obtained a similar trend as in PBS, showing a gradual increase in the current signal until the concentration of treated antibodies on the surface of AuNP-MB was 10 µg/mL. In addition, at a higher concentration (20 µg/mL), the current slightly decreased (data not shown). Hence, the concentration of capture antibodies for EV detection was optimized at 10 µg/mL, which showed that the number of immobilized antibodies on the AuNP-MB was maximized to capture any biomarkers as much as possible. Thus, the capture antibodies immobilized AuNP-MB through the avidin/biotin binding mechanism was eventually prepared for the EV assay.

### 3.4. Electrochemical Immunoassay on AuNP-MB Complex for EV Detection

To apply the developed EC biosensor platform using AuNP-MB for EV profiling, EVs extracted from the glioblastoma multiforme cell line (Gli36), a malignant form of brain cancer, were utilized (Appendix A). For the capture antibodies, anti-epidermal growth factor receptor (EGFR) antibodies were also used because EGFR is highly overexpressed on the EVs derived from the Gli36 cell line [46,47,48]. Additionally, EpCAM was used as a control because it is barely expressed on the Gli36-derived EVs [49,50,51,52]. Also, the expression level of EGFR and EpCAM on Gli36-derived EV was verified by Western blotting (See Appendix A). Thus, biotinylated anti-EGFR or anti-EpCAM antibodies were initially immobilized on the AuNP-MB complexes. Then, to label the captured EVs on AuNP-MB, anti-CD63 antibodies conjugated with ALP were utilized because CD63 is a well-known EV marker [53,54], which allows the sandwich assay for detecting GBM-derived EVs (Figure 1). To obtain the electrochemical signal for EV detection, chronocoulometry (CC) analysis was carried out at 0.6 V for 50 s, which was valid in our previous work due to its better reproducibility compared with CV analysis [55]. As shown in Figure 5, the biomarkers of Gli36-derived EVs were evaluated in the developed EC biosensor platform using the AuNP-MB complexes. The concentration of EV was 10^8^ ptcl/mL in the assay, which is able to handle the average EV concentration of 10^9^–10^10^ ptcl/mL present in body fluid [56,57,58,59,60]. As expected, the EGFR level was higher than the EpCAM level on the Gli36 GBM-derived EVs by showing higher charge values at 50 s at the applied potentials in both PBS and human plasma (1.45 $\times \;10^{-4}$ C for EGFR and 1.13 $\times \;10^{-4}$ C for EpCAM in PBS; 1.38 $\times \;10^{-4}$ C for EGFR and 8.87 $\times \;10^{-5}$ C for EpCAM in human plasma) (Figure 5). In addition, the EGFR signals showed a significant difference between the positive and negative samples. However, the EpCAM cases showed no difference between them. Additionally, to compare the case with just gold nanoparticles, we conducted a further experiment only using gold nanoparticles, and its results (see Appendix A)) are provided in the Appendix A. All the experimental conditions were the same as those used in the MB-based assay. The result for the EGFR biomarker showed that there was no difference between 0 and 10^8^ ptcl/mL in both PBS and plasma for the gold nanoparticles without MB. Therefore, the GBM-derived EVs in human plasma were well-evaluated, with the developed EC biosensor with AuNP-MB complexes showing higher sensitivity.

## 4. Conclusions

In this study, we developed a AuNP-coated MB-based electrochemical biosensor platform for profiling disease-associated EVs. First, the AuNPs conjugated on MB were well-fabricated via biological conjugation methods such as streptavidin/avidin and biotin-binding mechanisms. It was demonstrated that the AuNPs on the MB enhanced the electrochemical signal for electron transfer from the oxidation of AA from ALP–AAP reactions on a magnetic-beads-based platform. Therefore, with the AuNP-MB-based biosensor platform, we successfully evaluated disease-related biomarkers on GBM-derived EVs by detecting EVs in human plasma. Because the AuNP-MB complex shows higher sensitivity and is easy to use for bioassays, the AuNP-MB complex is a potential biosensor platform that can be used for increasing the sensitivity and simplicity of EV profiling immunoassay systems.

## Figures and Tables

**Figure 1 materials-15-06875-f001:**
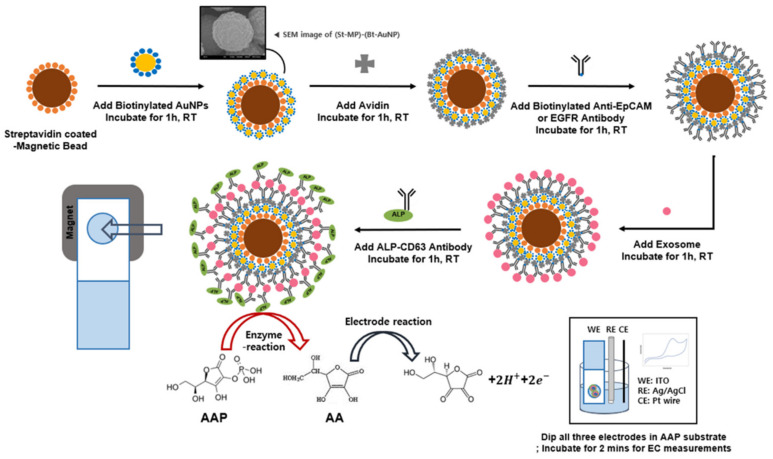
Schematics for AuNP-coated MB complex (AMB) preparation and electrochemical immunoassay on AMB.

**Figure 2 materials-15-06875-f002:**
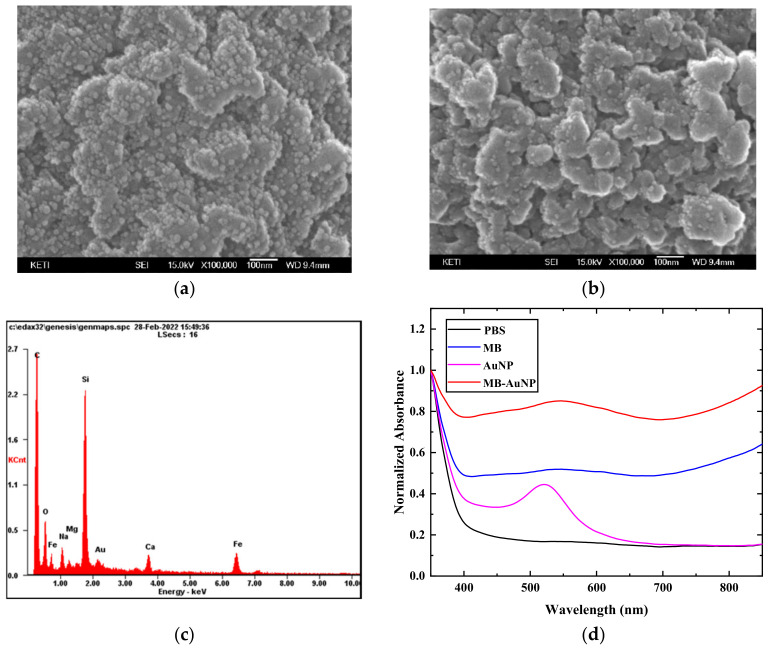
SEM/EDX analysis and UV–Vis absorption spectrum of AuNP-MB complex (AMB): (**a**) SEM image of AMB; (**b**) SEM image of MB; (**c**) EDX data of AMB; (**d**) UV–Vis absorption spectrum of the AMB, MB, and AuNP.

**Figure 3 materials-15-06875-f003:**
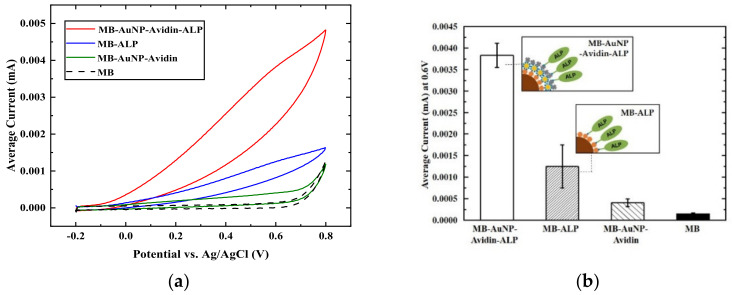
Analysis of signal enhancement of AMB: cyclic voltammetry (ranging from −0.2 V to 0.8 V) results of (**a**) AMB or MB labeled with ALP (red and blue lines, respectively). AMB or MB without ALP labeling were controls (green and dotted black lines, respectively); (**b**) current values for the tested samples at 0.6 V were recorded from Figure 3a; data = mean ± standard deviation, *n* = 3.

**Figure 4 materials-15-06875-f004:**
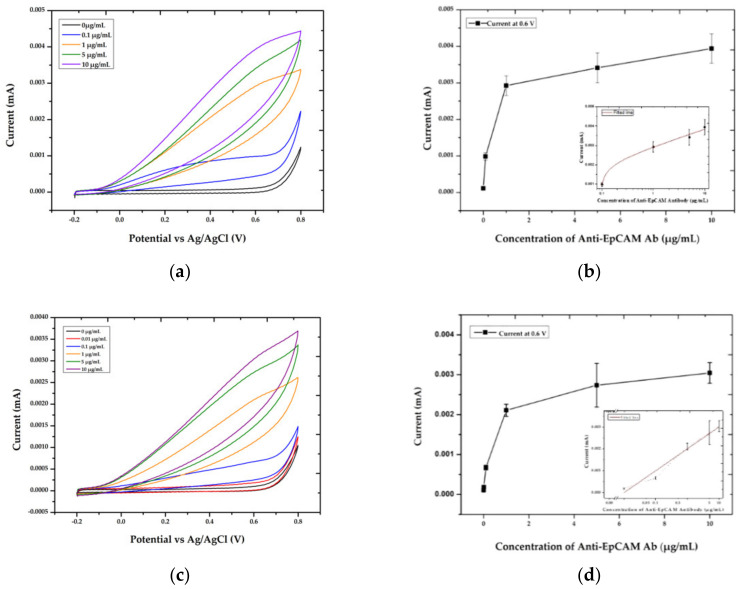
Cyclic voltammetry (CV) recorded (from −0.2 to 0.8 V) after an incubation period of 2 min at immunosensing electrodes treated with various concentrations of biotinylated anti-EpCAM mouse IgG antibody in PBS (**a**) and in 90% human plasma (**c**) in the developed AMB-based ITO platform. Anti-mouse IgG ab labeled with ALP was utilized for signaling. (**b**,**d**) Current values for the tested samples at 0.6 V were recorded in Figure 4a,c, respectively. Data = mean ± standard deviation, *n* = 3. As shown in the insets, all the data could be fitted using a linear line when plotting the X-axis in log-scale. The data were well-fitted with R^2^ of 0.99 in PBS for (**b**) and with R^2^ of 0.98 in 90% of human plasma for (**d**).

**Figure 5 materials-15-06875-f005:**
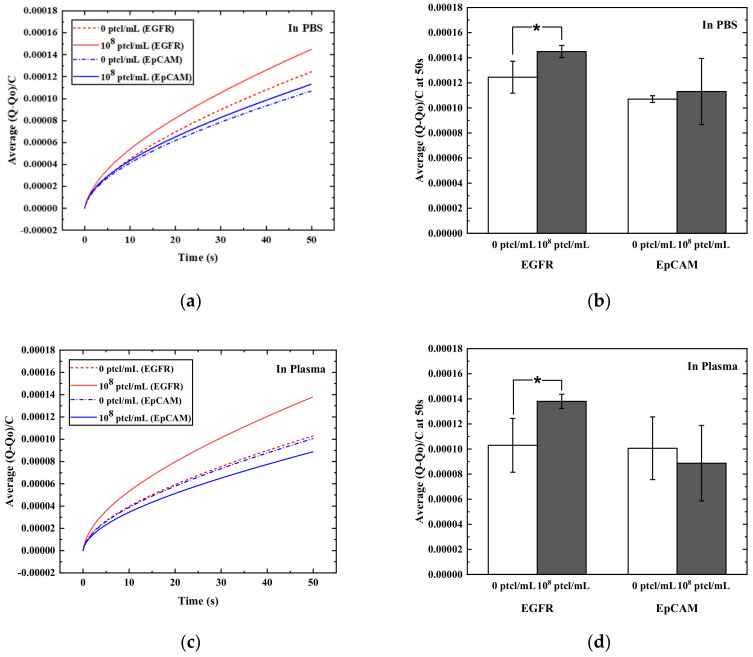
Chronocoulogram recorded (at 0.6 V) (**a**,**c**) and charge values at 50 s from Figure 5a,c are shown at 50 s after an incubation period of 2 min at immunosensing electrodes treated with different concentrations of EV and different capture antibodies (anti- EGFR or - EpCAM) in PBS (**a**,**b**) or in 90% of human plasma (**c**,**d**); Data = mean ± standard deviation, *n* = 3; * indicates a significant difference (*p* ≤ 0.05) between 0 and 10^8^ ptcl/mL of EVs with anti-EGFR antibodies.

## Data Availability

Data is contained within the article or Appendix A.

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
