# Peer review of "Gold-Nanoparticle-Coated Magnetic Beads for ALP-Enzyme-Based Electrochemical Immunosensing in Human Plasma"

_materials, 2022, doi:10.3390/ma15196875_

Round 1
Reviewer 1 Report
The work is well designed and the data supports the conclusion well.
1. The introduction could be expanded a bit and write in a more relavent form to magnetic microbeads-based biosensing. The relavent state of the art should be presented with clear advantages and disadvantages, and how they are related to the main target of this work.
2. I suggest that the author gives more thought to why magnetic microbeads are empolyed here. What is the main advantages since their charge transfer is not sufficient. What would the result be if we simply use gold nanoparticles? Would it yield similar sensing performance?
Author Response
"Please see the attachment."

Reviewer 2 Report
The reviewed paper describes the development of a magnetic-gold nanoparticles-based multi-sandwich platform for preconcentration exosomes from glioblastoma multiforme. Captured exosomes are labelled with alkaline phosphatase; the enzyme is used to split a phosphate monoester of ascorbic acid with the formation of free ascorbic acid. This is electroactive, and is used for the creation of a coulometric or voltammetric analytical signal. Despite the very complex preconcentration and determination procedure, the paper reports a valuable and successful attempt to determine cancer exosome. I strongly recommend accepting the paper. However, the paper is slightly unclearly written. More explanations should be added to the introduction and discussion. Therefore, minor revision is recommended.
Specific questions:
1. What is the source of the analytical signal (Fig. 5) at zero ptcl/ml?
2. Fig. 4 shows experimental results for Ep CAM antibody. Why was an analogical experiment not performed with EGFR antibody?
3. Which signal is more useful: that obtained by CV or coulometry?
Author Response
"Please see the attachment."
